# Exploration of the Antioxidant Effect of Spermidine on the Ovary and Screening and Identification of Differentially Expressed Proteins

**DOI:** 10.3390/ijms24065793

**Published:** 2023-03-17

**Authors:** Dongmei Jiang, Yongni Guo, Chunyang Niu, Shiyun Long, Yilong Jiang, Zelong Wang, Xin Wang, Qian Sun, Weikang Ling, Xiaoguang An, Chengweng Ji, Hua Zhao, Bo Kang

**Affiliations:** 1College of Animal Science and Technology, Sichuan Agricultural University, Chengdu 611130, China; 2Key Laboratory of Livestock and Poultry Multi-Omics, Ministry of Agriculture and Rural Affairs, Sichuan Agricultural University, Chengdu 611130, China; 3Farm Animal Genetic Resources Exploration and Innovation Key Laboratory of Sichuan Province, Sichuan Agricultural University, Chengdu 611130, China; 4Animal Nutrition Institute, Sichuan Agricultural University, Chengdu 611130, China

**Keywords:** spermidine, ovary, autophagy, antioxidant, proteomics

## Abstract

Spermidine is a naturally occurring polyamine compound that has many biological functions, such as inducing autophagy and anti-inflammatory and anti-aging effects. Spermidine can affect follicular development and thus protect ovarian function. In this study, ICR mice were fed exogenous spermidine drinking water for three months to explore the regulation of ovarian function by spermidine. The results showed that the number of atretic follicles in the ovaries of spermidine-treated mice was significantly lower than that in the control group. Antioxidant enzyme activities (SOD, CAT, T-AOC) significantly increased, and MDA levels significantly decreased. The expression of autophagy protein (Beclin 1 and microtubule-associated protein 1 light chain 3 LC3 II/I) significantly increased, and the expression of the polyubiquitin-binding protein p62/SQSTM 1 significantly decreased. Moreover, we found 424 differentially expressed proteins (DEPs) were upregulated, and 257 were downregulated using proteomic sequencing. Gene Ontology and KEGG analyses showed that these DEPs were mainly involved in lipid metabolism, oxidative metabolism and hormone production pathways. In conclusion, spermidine protects ovarian function by reducing the number of atresia follicles and regulating the level of autophagy protein, antioxidant enzyme activity, and polyamine metabolism in mice.

## 1. Introduction

Mammalian ovaries consist of follicles as basic functional units. The reproductive life span of the organism mainly depends on follicular development that maintains the primordial follicle pool in the cohort of follicles within the ovary [1]. The total number of ovarian follicles is determined early in life, and the depletion of the follicle pool leads to reproductive senescence [2]. Follicular atresia occurs in approximately 99% of the follicles in the mammalian ovary at various stages [3]. The total count of primordial follicles decreases with age due to ovulation and follicular atresia. Follicular atresia, a process of ovarian follicle degradation, mainly occurs via apoptosis, but recent studies also favor autophagy existence [1]. Generally, the proliferation and differentiation of granulosa cells lead to follicular maturation and ovulation, whereas apoptosis and degeneration of granulosa cells result in follicular atresia [4]. Ovarian granulosa cells are the only somatic cells that interact closely with oocytes and are extensively involved in primordial follicle recruitment, dominant follicle selection, steroid hormone secretion, and follicular atresia, and serve major roles in follicular development [5,6]. Polyamines are a class of polycationic fatty amines widely existing in the biological world, mainly including putrescine, spermidine and spermine. Natural polyamines themselves, especially spermidine and spermine, can act as reactive oxygen species (ROS) scavengers, thereby protecting cells from oxidative damage mediated by free radicals [7,8]. In recent years, a great quantity of in vitro and in vivo experimental studies have shown that polyamine depletion induces ROS accumulation and leads to cell damage, which in turn inhibits cell growth [9,10,11]. Exogenous spermidine can play antioxidative stress [12,13,14] and antiaging [15] roles by activating the autophagy pathway. Therefore, maintaining the stability of polyamine levels in the body is instrumental in maintaining the homeostasis of cellular redox. Spermidine is a polyamine widely present in animals, plants and microorganisms and can participate in the regulation of animal reproduction by mediating spermatogenesis in male animals, as well as follicle development, oogenesis and ovulation in female animals [16]. Apoptosis of granulosa cells is the initiating factor of follicular atresia, which is not conducive to the improvement of animal productivity and is also associated with premature ovarian failure [5]. It has been proved that oxidative stress can induce apoptosis. Spermidine is an antioxidant [17] that is closely related to ovarian function. 

In addition, studies in recent years have shown that spermidine can reduce ROS levels and delay cell aging in humans and other models, such as yeast, Drosophila and nematode models [18]. Adding spermidine or feeding spermidine-rich food can significantly improve the lifespan of rats [15]. Spermidine and spermine themselves protect E. coli and mammalian cells from oxidative damage caused by H_2_O_2_ [10,19]. Using microarrays to compare yeast (spermine-deficient) mutants containing lower levels of spermidine with yeast mutants supplemented with spermidine, spermidine altered the expression of 500 genes by more than 2-fold, including several genes involved in the oxidative stress response (*HSP12, GAD1, GPX2*, etc.) [20], suggesting that spermidine can regulate a variety of antioxidant enzymes at the transcriptional level. Studies have found that spermidine can activate the Keap1-Nrf2-ARE antioxidant signaling pathway and then mediate heme oxygenase-1 (HO-1) and reduce coenzyme (NAD(P)H quinone oxidoreductase 1. Endogenous antioxidant enzymes such as NQO 1 and catalase (CAT) are involved in regulating the body’s antioxidant function [21,22]. In addition, a recent study showed that spermidine could manifest an antioxidative stress role by increasing the level of glutathione (GSH) and reducing the content of malondialdehyde (MDA) in rat brain tissue, reducing lipid peroxidation [23]. Spermidine can also participate in free radical scavenging activities, reducing age-related oxidative protein damage and ROS overproduction [24,25]. To date, the research on spermidine’s antioxidative stress has mainly focused on neural tissue, while research on spermidine’s involvement in regulating the antioxidant function of the female ovary and further affecting its reproductive and production performance is relatively rare, and the corresponding mechanisms remain unclear. 

Under normal physiological conditions, autophagy maintains the physiological activities of cells by specifically degrading damaged or redundant organelles, so autophagy is an important cytoprotective mechanism. However, excessive autophagy is detrimental. Studies have shown that nonapoptotic forms of programmed cell death (PCD), such as autophagy, may also be involved in the process of follicular atresia [26,27,28]. More autophagosomes can be detected in the granulosa cell layer of atretic follicles [29], and ROS, as a byproduct of aerobic metabolism in the body, can induce oxidative stress and cell damage when accumulated in large amounts [30]. Animal reproductive activities often require large amounts of energy and nutrients, which also lead to the production and enrichment of ROS [31]. Numerous studies have shown that spermidine can enhance cellular function by activating autophagy. Spermidine lost the ability to scavenge ROS after the knockout of the autophagic protein Beclin 1 [18]. It is suggested that spermidine can induce autophagy and inhibit follicular atresia in mice.

In recent years, proteomics technology has provided technical support for ovarian-related research. For example, comparing the differences in serum protein expression levels between patients with polycystic ovary syndrome (PCOS) and non-PCOS patients is helpful for finding potential serum markers [32]. Additionally, integrative metabolomics and proteomics were used to highlight altered polyamine pathways in lung adenocarcinomas [33]. Combined single-nucleus sequencing (snRNA-seq) and proteomic analysis revealed a direct anti-inflammatory effect of spermidine on glial cells in a mouse model in an autophagy-dependent manner [34]. In this study, mice were fed with exogenous spermidine for a period of time. We investigated the effect of spermidine on the antioxidation of ovarian function by detecting the histological changes of the ovary, the expression level of autophagy protein and follicular development-related protein, the activity of antioxidant enzymes, the content of polyamines and the expression of polyamine metabolism genes. Further, the differential proteins related to antioxidation and autophagy of spermine were mined by using the LFQ proteomics technology, and the protein expression profile of spermine on ovarian function was screened and analyzed, providing clues for further research.

## 2. Results

### 2.1. Effects of Spermidine on Water Intake, Body Weight and Ovarian Index in Mice

After feeding female ICR mice with 3 mmol L^−1^ spermidine for 3 months, it was found that the weight of the mice increased steadily but did not differ significantly between the two groups (Figure 1A). This shows that spermidine has no harmful effect on mice. The daily food intake increased significantly but did not lead to significant differences in daily weight gain or water intake (Figure 1B,D,E), and spermidine had no effect on the ovarian index of mice (Figure 1C).

### 2.2. Effects of Spermidine on Ovarian Histomorphology and Follicular Development in Mice

We found that the boundary between the mouse ovarian cortex and medulla was clear: round or oval follicles with different sizes were visible, the number of oocytes did not increase significantly, there was no obvious interstitial hyperplasia in the medulla, and the shape and number of primordial follicles, growing follicles and mature follicles in the cortex were not significantly different from the control group (Figure 2A). The number of atresia follicles and the extent of corpus luteum in the spermidine treatment group decreased significantly (Figure 2B).

To verify the effect of spermidine on mouse follicular development, we quantified proteases related to ovarian steroid hormone synthesis. Many studies have demonstrated that steroid hormones play important roles in regulating follicular growth and atresia. Cytochrome P45017A1 (CYP17A1), 3 β-hydroxysteroid dehydrogenase (3β-HSD), and 17 β-hydroxysteroid dehydrogenase (17β-HSD) and other enzymes are expressed in growing follicles, and the corpus luteum, and their main function is to convert estrogen into E2, thereby affecting follicular development. The results showed that the protein expression of CYP17A1 and HSD3B2 in mouse ovaries treated with spermidine was significantly increased (Figure 2C,D)

### 2.3. Effects of Spermidine on Polyamine Content and Expression of Key Metabolic Genes in Mouse Ovaries

The spermine content in the treatment group was significantly higher than that in the control group, which was approximately 1.2 times that of the control group. There was no significant effect of putrescine and spermidine content in ovaries (Figure 3A). The relative expression of the polyamine oxidase (*APAO)* gene was significantly upregulated in the treatment group. The relative expression of the spermidine synthase *(SPDS)*, spermine synthase *(SPMS)* and spermine oxidase *(SMO)* genes was significantly lower than in the control group. The relative expression of the ornithine decarboxylase *(ODC)* and N’-spermidine/spermine acetyltransferase *(SSAT)* genes was not significantly different (Figure 3B).

### 2.4. Spermidine Activates Antioxidant Enzyme Activity to Protect the Ovary

Further exploration of the antioxidant function of spermidine on the mouse ovary showed that the total antioxidant capacity of the mouse ovary was significantly increased to approximately 1.62 times that of the control group (Figure 4B). Both SOD and CAT enzyme activities were significantly higher than those of the control group, and they were 2.50 times and 1.67 times those DEPs of the control group, respectively (Figure 4A,C). The ovarian MDA level of the spermidine-treated group was significantly lower than that of the control group, which was 0.47 times that of the control group (Figure 4E). Feeding spermidine had no significant effect on the enzymatic activity of mouse ovary GSH-px (Figure 4D). 

### 2.5. Spermidine Induces Ovarian Autophagy in Mice

SPD antioxidant effect is achieved through the rescue of autophagic flux, but there is a lack of research in the context of the ovary. Therefore, we determined the expression and localization of autophagy proteins in ovaries after spermidine treatment. The positive products of the autophagy marker proteins LC3 and p62 were mainly distributed in the cytoplasm and mainly concentrated in the granulosa cell layer of antral follicles. After feeding with spermidine, the LC3 protein expression signal in mouse ovarian tissue was enhanced, while the p62 protein expression signal was weakened (Figure 5A). The protein expression levels of Beclin 1 and LC3II/I in the ovaries of the treated mice were significantly increased and were 1.93 times and 1.83 times those of the control group, respectively. Compared with the control group, the expression of p62 protein was significantly decreased to 0.53 times that of the control group (Figure 5B,C).

### 2.6. Protein Sample Consistency Test and Protein Identification

Next, with the purpose of investigating the potential mechanisms of spermidine action against ovarian oxidative stress and induced autophagy, we conducted non-label-free proteomic sequencing to screen and identify the key proteins that confer protective effects on the mouse ovary. Sample quality and repeatability are important for protein sequencing. The results of the protein quantitative principal component analysis showed that the quantitative repeatability of the aggregation degree between repeated samples was good, the difference between the two groups was large (Figure 6A), the RSD box axis was relatively centered (Figure 6B), and the Pearson correlation coefficient between samples was close to one. The samples were shown to be reproducible (Figure 6C), and the samples were available. The distribution of peptide lengths (7–20 amino acids) identified by mass spectrometry met the quality control requirements (Figure 6D). A negative correlation between the molecular weight of the protein and the coverage was observed. To achieve equal coverage, more peptides must be identified for a large protein (Figure 6E).

### 2.7. Screening of Differentially Expressed Proteins

Next, we screened 6545 DEPs by nonstandard quantitative proteomic sequencing, of which 5498 proteins could be quantified (Figure 7A). Compared with the control group, the 3 mmol·L^−1^ spermidine-treated group had 681 differentially expressed proteins and 424 upregulated proteins, including tumor protein p53 (P53), nuclear receptor subfamily 5 group A member 1 (Nr5a1), BCL2-related ovarian killer (Bok), and BH3 domain apoptosis-inducing protein (Bid). There were 257 downregulated proteins, including xanthine dehydrogenase (XDH) and thioredoxin-interacting protein (TXNIP) (Figure 7B and Table 1). These proteins are mainly involved in functions such as “steroid synthesis, apoptosis, autophagy and oxidative stress”.

### 2.8. Functional Classification, Subcellular Structure Localization Classification, and COG/KOG Functional Classification of Differentially Expressed Proteins

Most of the DEPs have transporter, transducer, binding and catalytic activities and participate in cellular biological processes (Figure 8A). We classified the proteins identified in this study according to their cellular components based on GO analysis. The DEPs were mainly distributed in the nucleus, cytoplasm, extracellular space, mitochondria and plasma membrane of the mouse ovary. (Figure 8B). Signal transduction mechanisms (enrich 76 proteins); lipid transport and metabolism (enrich 45 proteins); replication, recombination and repair (enrich 21 proteins) (Figure 8C).

### 2.9. Functional Enrichment Analysis of Differentially Expressed Proteins

In terms of biological processes, DEPs are involved in steroid biosynthesis, cholesterol metabolism, lipid homeostasis and transport, and cellular hormone metabolism (Figure 9A). In terms of cellular components, the DEPs mainly play roles in protein extracellular matrix, nuclear chromosome parts and telomeric regions, extracellular matrix component, and chromosomal regions (Figure 9B). In terms of biological functions, the main functions of DEPs, including sterol transporter activity, copper ion binding, cholesterol transporter activity, and especially steroid dehydrogenase activity, were more obvious (Figure 9C). KEGG analysis revealed that DEPs were enriched in signaling pathways such as DNA replication, ovarian steroidogenesis, steroid biosynthesis, and glycerol lipid metabolism (Figure 9D).

### 2.10. Protein Domain Enrichment

In order to study the physiological functions of proteins, we enriched protein domains. Protein domains were mainly enriched in the MCM domain, MCM OB domain, MCM N-terminal domain; EGF-like domain, extracellular; START domain; alkaline phosphatase-like, core domain; AMP-dependent synthase/ligase, etc. (Figure 10). 

### 2.11. Analysis of Protein Interaction Network

A total of six DEPs were screened, namely, XDH, TXNIP, p53, Nr5a1, Bok and Bid. XDH is an enzyme that catalyzes purine metabolism, which may be related to oxidative stress. TXNIP is also related to oxidative stress, and there seems to be some connection to excessive autophagy. p53 is a cell cycle factor that participates in the cell block. Nr5a1 is a transcription factor for steroidogenesis. Bid and Bok are both proapoptotic proteins and are involved in inducing apoptosis (Figure 11).

## 3. Discussion

Putrescine, spermidine and spermine are interconverted through the polyamine metabolic system to maintain the stability of the body’s polyamine pool, and this process requires the participation of a variety of polyamine metabolizing enzymes [35]. In our experiments, after feeding with 3 mmol·L^−1^ spermidine for 3 months, the expression of the *ODC* gene in the ovary of mice was statistically insignificant, and the level of putrescine in the ovary did not change, indicating that 3 mmol·L^−1^ spermidine did not affect the synthesis of putrescine in the ovary of mice. Furthermore, the expression of the *SPDS*, *SPMS* and *SMO* genes in the ovaries of mice in the spermidine group was significantly decreased. *APAO* had the opposite effect (the levels of spermidine and putrescine were unchanged, and the level of spermine was significantly increased). It is speculated that a portion of the ingested spermidine is converted into spermine for storage, and part of the ingested spermidine forms N 1-acetyl-spermidine under the action of *SSAT* to maintain spermidine in the ovary. In addition, the decreased expression of *SMO* is also one of the reasons for the increased spermine level in the ovary. Natural polyamines themselves can bind to negatively charged biological macromolecules such as nucleic acids, proteins, and biofilms, thereby exerting the functions of antioxidative stress and scavenging ROS. However, excess polyamines are catabolized by various amine oxidase enzymes, which produce many reactive aldehydes (such as acrolein) and H_2_O_2_, which are toxic and detrimental to proteins, DNA in mammalian cells, and other cellular components that are susceptible to oxidative damage [21,36]. Therefore, how to effectively utilize the antioxidant capacity of polyamine while avoiding oxidative stress caused by polyamine metabolism has become a major problem in the study of polyamine functions. In addition, both spermidine and spermine can be degraded by acetylpolyamine oxidase, and spermine oxidase can directly degrade spermine. While both pathways can produce toxic products, including H_2_O_2_, oxidation by spermine oxidase is more likely to cause damage because spermine oxidase is present in the nucleus and cytoplasm, while acetylpolyamine oxidase is a peroxidant. The bioenzyme itself has a protective effect on cells [37]. In this experiment, the expression of *SMO* in the ovary decreased significantly after spermidine feeding. Our experimental results are consistent with previous results [38]. We speculate that exogenous spermidine can regulate polyamine catabolism by inhibiting the expression of SMO at the transcriptional level and avoiding oxidative damage. Therefore, spermidine can maintain the stability of the polyamine pool by regulating the genes encoding polyamine-metabolizing enzymes.

Studies have shown that many pathological phenomena in the body are closely related to lipid peroxidation caused by free radicals [30,39]. In recent years, oxidative stress has taken an important part in maintaining the normal physiological function of the female reproductive system and the pathogenesis of reproductive diseases such as polycystic ovary syndrome [40]. CAT is an important enzyme in the body’s enzymatic antioxidant system that scavenges H_2_O_2_ and indirectly inhibits lipid peroxidation and membrane damage. The activity of CAT can indirectly reflect antioxidant capacity [41]. GSH-Px is a known antioxidant enzyme that can convert peroxides and hydroxyl radicals into nontoxic forms [42]. MDA is not only an important product of lipid peroxidation but can also cause intramolecular and intermolecular cross-linking of proteins by reacting with free amino acids of proteins, resulting in cell damage [43]. As a secondary product of lipid peroxidation, MDA reflects the occurrence of ROS and can be used as an indicator of membrane damage [44,45]. Oxidative damage is usually accompanied by decreased activities of CAT, SOD and GSH-px enzymes and an increase in the level of MDA [46,47]. Jantaro et al. [48] indicated that the exogenous addition of spermidine could significantly alleviate the increase in ROS levels and lipid peroxidation caused by ultraviolet irradiation in algal cells and had a protective effect on the enzyme activities of SOD and CAT in cells. Recent studies have shown that during cisplatin-induced nephrotoxicity, spermine supplementation can effectively inhibit oxidative stress and nitric acid stress and reduce DNA damage and lipid peroxidation, thereby preventing cisplatin-induced renal tubular necrosis [49]. During follicle development, spermidine is crucial in the regulation of ovarian granulosa cells. External stress may cause apoptosis of ovarian granulosa cells, resulting in follicular atresia and affecting the reproduction of female animals. Our results are consistent with previous findings [50,51] that feeding with 3 mmol·L^−1^ spermidine significantly increased the T-AOC/SOD and CAT enzyme activities of mouse ovarian tissue without affecting the daily weight gain and ovarian organ index of mice. It significantly reduced the MDA content and decreased the level of ovarian lipid peroxidation. It also significantly reduced the number of atretic follicles in mice. Moreover, the protein expression of CYP17A1 and HSD3B2 in ovarian tissue was increased, thereby promoting follicle development. In conclusion, adding spermidine to drinking water can significantly reduce the atresia rate of follicles in mice and significantly improve the antioxidant capacity of mouse ovaries by increasing the activities of various antioxidant enzymes and reducing lipid peroxidation.

It was found that autophagy levels were significantly increased in various tissues (heart, liver and muscle) from 4 h to 24 h by acute intraperitoneal injection of spermidine in mice supplemented with spermidine in drinking water. Changes in mitochondrial structure and function are an initial factor in cellular senescence [52,53], and studies have shown that spermidine can promote cell-dependent selective degradation (mitophagy) in human fibroblasts and mouse neuroblastoma (N_2_a) cells. This contributes to mitochondrial health and function, which in turn has anti-aging effects [39,54]. Recent studies have shown that spermidine can also inhibit the degeneration of the intervertebral disc by inducing autophagy and then reducing the apoptosis of nuclear myeloid cells [55]. In the field of reproduction, the regulation of autophagy and antiaging effects of spermidine have rarely been reported. Consistent with existing results, we found that spermidine also induces autophagy in mouse ovaries, but the exact mechanism remains unclear. Feeding spermidine significantly increased the expression of the key autophagy proteins Beclin 1 and LC3II in the ovary; at the same time, the expression of p62 protein was significantly decreased, indicating that 3 mmol·L^−1^ spermidine feeding can induce autophagy. At the same time, the immunohistochemistry results showed that the positive products of the autophagy marker proteins LC3 and p62 were mainly distributed in the cytoplasm and concentrated in the granulosa cell layer of antral follicles. Phagocytosis marker proteins were specifically distributed in the ovary, mainly in granulosa cells. Properly increasing the level of autophagy can help to improve oxidative stress and alleviate neural deafness due to aging or drugs [56]. The above studies show that autophagy has a positive impact on improving the antioxidant capacity of the mouse ovary.

The specific mechanism by which spermidine induces autophagy in mouse ovaries to protect against oxidative stress is unclear. Therefore, our research group used proteomic sequencing to screen differentially expressed proteins and provide theoretical support for subsequent experiments. In this study, proteomic methods revealed 681 DEPs in the ovarian tissue of spermidine drinking water-fed mice, including 424 upregulated proteins and 257 downregulated proteins, including p53, Nr5a1, Bok, Bid, XDH and TXNIP, etc. These proteins are mainly involved in functions such as “cell cycle, steroid synthesis, apoptosis, autophagy and oxidative stress”. The expression of several ribosomal proteins and histones was increased, and the upregulated proteins are known to be involved in apoptosis and other processes, indicating that spermidine can promote protein synthesis and apoptosis in ovarian tissue. At the same time, the expression levels of several ribosomal proteins and molecules related to oxidative metabolism were downregulated, and the downregulated proteins are known to be involved in energy metabolism, signal transduction, RNA processing and modification, transcription, lipid transport and metabolism, and other processes. The treatment has an inhibitory effect on some functions of ovarian tissue. GO annotation and functional enrichment analysis showed that the functions of most of the DEPs were related to cell binding, enzyme catalytic ability and molecular function regulation, indicating that spermidine acts on ovarian granulosa cells and through the regulation of related molecular functions, it can change the interactions among ovarian granulosa cells. In our research, protein domains were mainly enriched in the MCM domain, MCM OB domain, MCM N-terminal domain; EGF-like domain, extracellular; START domain; alkaline phosphatase-like, core domain; AMP-dependent synthase/ligase, etc. The MCM protein family is an important part of the DNA prereplication complex and plays an important role in the DNA replication initiation process and DNA damage repair. EGF may regulate reproductive function by controlling the secretion of transferrin; aging can affect the transcription of EGF and EGFR. START and AMP-dependent enzymes are closely related to ovarian steroidogenesis. The KEGG pathway enrichment results showed that the DEPs were involved in multiple cell signaling pathways, including the p-ERK, JNK, p53, TNF, TXNIP and DR signaling pathways. Some research has shown that the p53 [57] signaling pathway is involved in the regulation of ovarian granulosa cell cycle arrest and apoptosis caused by oxidative stress, and these signaling pathways play important roles in regulating ovarian granulosa cell function and follicle development. 

An appropriate level of autophagy in the body is a prerequisite for the body to maintain normal physiological functions. Insufficient or excessive autophagy can lead to cell death. TXNIP is a widely expressed protein that is induced by various cellular stresses, including oxidative stress and apoptosis [58]. TXNIP is a pro-oxidative protein that negatively regulates thioredoxin activity and its antioxidant function [59]. Gao et al. found that TXNIP significantly reduced the expression of LC3 and p62 proteins during myocardial ischemia/reperfusion (I/R), indicating that TXNIP is an autophagy regulator [60]. Several studies have shown that regulation of developmental and DNA damage response 1 (REDD1) can be bound and stabilized by TXNIP. TXNIP increases autophagosome formation during I\/R by upregulating REDD1 and inhibits autophagosome clearance in vivo by ROS induction [58,61]. TXNIP-REDD1 plays a role in excessive myocardial autophagy, which is a new way to regulate autophagy [62]. Previous studies found that TXNIP can regulate the autophagy of Mueller cells [63] and tubular cells [64] by inhibiting the activation of mTOR. Consistent with this, in Caco-2 cells, the bidirectional effect of TXNIP on autophagic flux may also regulate the change in SQSTM1/p62 protein [65]. In our research, we found by proteomics that spermidine downregulated the protein expression of TXNIP in mouse ovaries. In addition to its role in mediating oxidative stress, TXNIP is also a factor closely related to the reproductive system. Oxidative stress induces excessive secretion of androgens, which in turn induces IR in PCOS [66]. Localized IR in the ovary may lead to abnormal follicular development, ovulatory disorders and hyperandrogenism, resulting in reproductive disorders. The levels of testosterone (T), Luteinizing hormone (LH), Estradiol (E2), and the mRNA levels of *TXNIP* and insulin-like growth factors (*IGF-1*) in ovarian tissue decreased after TXNIP was knocked down, and the level of FSH increased. This result indicated that inhibiting the expression of TXNIP could reduce the expression of ovarian function-related hormones. The results of Illumina sequencing and qRT-PCR showed that the *CYP19A1* and *TXNIP* genes might take part in promoting the development of bovine follicles and ultimately cause follicle ovulation [67]. *TXNIP* is highly expressed in bovine cumulus cells [68], porcine oocytes [69] and porcine oviduct epithelial cells [70]. This gene is also important in the maturation of mouse oocytes [71]. These results provide a potential reference for the study of mammalian follicular mechanisms and provide a possible new genetic marker for the study of the pig granulosa cell cycle [72]. These data suggest that future research directions will involve the exploration of the specific functions of TNXIP in the ovary. Studies have shown that excessive oxidative stress can lead to ovarian granulosa cell damage [73], which in turn induces granulosa cell apoptosis, leading to follicular atresia [74]. Spermidine has been reported to inhibit reactive oxygen species and may act as an endogenous reactive oxygen species scavenger [10]. This result suggests that spermidine may play an antioxidant role by downregulating the expression of TXNIP in ovarian granulosa cells to regulate autophagy. These results provide new insights into the future exploration of spermidine-induced autophagy to exert anti-oxidation and thus protect mammalian ovarian function. However, the specific mechanism of the screened key functional proteins needs further verification.

## 4. Conclusions

Our current findings suggest that spermidine regulates the activity of antioxidant enzymes and the expression level of autophagy proteins, and autophagy activated by spermidine plays an antioxidant role in preventing follicular atresia. Using proteomics, we found that it is mainly involved in steroid hormone production, oxidative stress, lipid metabolism, autophagy and apoptosis, and the main proteins are Nr5a1, XDH, P53, LDLR, TXNIP, Bok and Bid. In summary, spermidine protects ovarian function by reducing the number of atresia follicles and regulating the level of autophagy protein, antioxidant enzyme activity, and polyamine metabolism in mice.

## 5. Materials and Methods

### 5.1. Experimental Animals and Sample Collection

Seven-week-old healthy female ICR mice (SPF-grade experimental mice purchased from Chengdu Da Shuo Laboratory Animal Co., Ltd., Chengdu, China) were selected and kept in separate cages, and the mice were fed ad libitum. The mice were randomly divided into a control group (normal drinking water) and a spermidine group (3 mmol ·L^−1^ spermidine solution instead of drinking water), with 20 mice in each group. During the feeding process, the water intake, cleaning of the drinking water bottle, and replacement of the drinking water were recorded every three days, and the animals were weighed and recorded every 15 days. After 3 months, the reproductive cycle was identified by the vaginal smear method, and mice in oestrus were selected. The mice were killed by cervical dislocation, and the bilateral ovaries were quickly collected, washed with 0.9% normal saline, dried with filter paper, and placed on an electronic balance. Some of the ovarian tissue samples were fixed with 4% paraformaldehyde for HE staining, some were sequenced by proteomics, and the rest were stored in a −80 °C freezer for later use. 

This experiment was completed in the College of Animal Science and Technology, Sichuan Agricultural University. All feeding and killing requirements were in line with the animal operation code of Sichuan Agricultural University and approved by the Welfare Administration Committee (DKY-B2019202011).

### 5.2. Main Reagents and Kits

Putrescine standard, spermidine standard, spermine standard and 1,6-hexanediamine were purchased from Sigma (St. Louis, MO, USA); Protein Maker, ECL chromogenic solution and PVDF membrane were purchased from BIO-RAD Biological Company; PBS, PMSF, RIPA, primary antibody diluent and secondary antibody diluent were purchased from Beyotime Co., Ltd. (Haimen, China); chromatographic methanol was purchased from Bio-Rad Yu (Fisher Scientific, Waltham, MA, USA); SYBR Green Supermix was purchased from TaKaRa (Kusatsu, Japan); PMSG was purchased from Ningbo Sunshine Biological (Wuxi, China); perchloric acid, sodium hydroxide, concentrated hydrochloric acid, benzoyl chloride, anhydrous ethanol and all other reagents were analytically pure and purchased from Ruijinte (Chengdu, China).

### 5.3. Analysis Software 

The proteins obtained by liquid chromatography-mass spectrometry were analyzed by means of bioinformatics. The tools and websites used for the analysis are shown in Table 2.

### 5.4. Morphological Detection of Ovarian Tissue

The ovarian tissue was fixed in 4% paraformaldehyde for 24 h, after which the tissue was removed, dehydrated in a series of alcohols with different concentrations, embedded in paraffin, and sectioned. Next, hematoxylin-eosin staining was performed. The slides were sealed with neutral gum and dried in an oven at 37 °C. The changes in the tissue morphology of the ovary, the size of the follicles, and the granulosa layer of the follicles were observed under an optical microscope, and the follicles of different grades were counted.

### 5.5. Immunohistochemical Detection of Autophagy-Related Proteins in Ovarian Tissue

First, the tissue was embedded and sectioned, the sections were dewaxed and then stained, and the sections were infiltrated with 3% methanol and hydrogen peroxide for 9 min, washed three times with PBS, and immersed in 0.01 mol/L citrate buffer (PH 6.0). After heating for 5 min, the slices were allowed to cool completely, and then the slices were washed 2–3 times with PBS. At room temperature, goat serum blocking solution was used to block the slices for 20 min. The primary antibody was diluted according to the antibody instructions, added dropwise to the slice, and incubated at 4 °C overnight. The secondary antibody was incubated at 37 °C for 30 min. The slices were washed three times with PBS for 4 min each time. After DAB color development at room temperature, the slices were placed under a microscope, and the reaction was observed and rinsed with distilled water after approximately 2–3 min. Counterstain was performed, followed by dehydration and xylene clearing. Finally, the sections were mounted with neutral gum. The sections were observed microscopically, and images were acquired for subsequent analysis.

### 5.6. Antioxidant Index and Lipid Peroxidation Levels Detection

An appropriate amount of ovarian tissue sample was added to prepare the solution. Then, a hand-held electric homogenizer was used to homogenize the samples on ice, and the supernatant was collected as the sample to be tested. The protein concentration was determined with a BCA protein concentration kit for quality control for subsequent detection. The working solution was prepared according to the instructions of the kit, the sample was diluted, the 96-well plate was gently shaken to fully mix the solution, the plate was incubated in an incubator at 37 °C, and the absorbance of the sample was detected at 450 nm. The enzyme activity was calculated according to the instructions of the kit {CAT (Beyotime, S0082, Shanghai, China), SOD (Beyotime, S0101S, Shanghai, China), GSH-Px (Beyotime, S0056, Shanghai, China), total antioxidant capacity (T-Ait isOC, Beyotime, S0119, Shanghai, China) and MDA (Beyotime, S0131S, Shanghai, China)}.

### 5.7. Determination of Polyamine Content in Mouse Ovarian Tissue by High-Performance Liquid Chromatography

Approximately 0.1 g of mice ovary tissue was added to an internal standard and 1 mL 5% HClO4 grinding sample and placed on ice for subsequent detection. Vortex vibration and ultrasonic crushing were performed for 10 min, followed by centrifugation and supernatant extraction, which was repeated 1–2 times. Aliquots of 2 mL 2.5 mol/L NaOH and 7 μL benzoyl chloride were added to the above homogenate solution, vortexed and mixed, then derivation was performed in a water bath without light, with the derivatization solution adjusted to pH 7.0 with 6 mol/L HCl. The samples were activated with chromatographic methanol and ultrapure water in advance, and the polyamines were completely filtered and separated, washed with 15 mL double distilled water and 15 mL 15% chromatographic methanol, and dried. Then, 0.5 mL chromatographic methanol was used to elute the components to be tested. A 0.22 μm filter was used to filter the prepared sample solution, and the filtered sample solution was detected by HPLC. The detection conditions were as follows: the mobile phase was methanol: water (62:38, *v*/*v*), the detection wavelength of the UV detector was 229 nm, and the column temperature was 25 °C. The peak time and peak area of three polyamines and internal standard in each sample were obtained by liquid chromatography, and the concentration of each polyamine was calculated according to the standard curve.

### 5.8. Real-Time Fluorescence Quantitative PCR Detection

After grinding the ovarian tissue with liquid nitrogen, total RNA was extracted according to the instructions of the RNAiso Plus kit (TaKaRa, 9109, Dalian, China), according to the instructions of the reverse transcription kit (TaKaRa, RR047A), and the total RNA sample was reverse transcribed into a cDNA template. Primer information is as follows (Table 3. The mixed system {TB Green Premix EX Taq II(TaKaRa, RR820A, Dalian, China) 5.0 μL; PCR upstream primer, 0.2 μL; PCR downstream primer, 0.2 μL; cDNA 0.5 μL; dd H2O, 4.1 μL}. The reaction conditions were as follows: pre-denaturation at 95 °C for 3 min; 39 cycles of denaturation at 95 °C for 10 s, annealing at 57–63 °C for 30 s, and extension at 72 °C for 30 s (fluorescence collection); holding at 95 °C for 10 s. The β-actin gene was used as an internal reference gene. Three repetitions were performed for each sample, and 2^−ΔΔ^ was used to calculate the relative expression of mRNA by the Ct method.

### 5.9. Western Blot Detection of Protein Expression

After the ovarian tissue was ground with liquid nitrogen, 0.1% protease inhibitor (PMSF) and protein lysis buffer (PIPA) were added and centrifuged at 4 °C at 12,000 r/min for 10 min, and the protein of the sample was detected according to the BCA protein concentration detection kit (Beyotime, P0009) after concentration and denaturation. After preliminary separation by SDS-PAGE, the proteins were transferred to PVDF membranes for sealing, followed by 4 °C, overnight incubation of primary antibody {1:1000, rabbit polyclonal antibody CYP17A1 (Abcam, ab125022, Cambridge, UK), HSD3B2 (Abcam, ab191515, UK), p62/SQSTM1 (CST, 23214S, US), LC3B (Sigma, L7543-100UL, US), rabbit monoclonal antibody Beclin 1 (Abcam, ab207612, UK), mouse monoclonal antibody GAPDH (Trans Gen Bio, HC301, Beijing, China) and rabbit monoclonal antibody β- Tubulin (ABclonal, A12289, Wuhan, China)}. The membrane was incubated with secondary antibody {1:1000, goat anti-rabbit secondary antibody (Beyotime, A0208), goat anti-mouse secondary antibody (Beyotime, A0216)} for 60 min at 37 °C. An ECL reagent kit was used for color development, exposed with an AI 600 imager, and analyzed with image analysis software (ImageJ 1.8.0.112, Bethesda, MD, USA).

### 5.10. Protein Extraction and Consistency Test of Repeated Samples

After the mice were fed with 3 mmol·L^−1^ spermidine drinking water for three months, ovarian tissue was collected. The samples were ground in liquid nitrogen, lysis buffer (inhibitor) was added, each group of samples was lysed by ultrasonication, and the supernatant was collected after centrifugation. SDS-PAGE was used for quality control. The Pearson correlation coefficient (Pearson) was used to measure the degree of linear correlation between the two groups of data; the relative standard deviation (RSD) was used to calculate the ratio of the standard deviation to the arithmetic mean of the measurement results, which reflected the degree of dispersion of the data; the principal component analysis (PCA) method extracts the key components in the sample data to effectively distinguish the samples, intuitively reflects the relationship between the samples, and assesses the consistency and repeatability of samples.

### 5.11. Liquid Chromatography-Mass Spectrometry (LC-MS) Analysis

After dissolution with liquid chromatography mobile phase A ((*v*/*v*) formic acid in water), the Nano Elute ultrahigh-performance liquid system was used for separation. The peptides were separated by an ultrahigh-performance liquid phase system, injected into the capillary ion source for ionization, and then analyzed by a tims-TOF Pro mass spectrometer.

### 5.12. Database Search and Bioinformatics Analysis

MS/MS data were retrieved using Max Quant (v. 1.6.6.0). The nonstandard quantitative calculation method was used to calculate the nonlabeled quantitative intensity of protein in each sample, and the relative quantitative value of each sample was obtained. The average value was obtained, and the final differential expression amount was calculated. Differential proteins were screened based on the *p*-value and differential expression of the original data, and GO annotation was used to classify the differential proteins according to cellular components, molecular functions or physiological processes. At the same time, the protein pathways were annotated using the KEGG annotation tool KASS, and the KEGG mapper matched the differentially expressed proteins to the corresponding KEGG pathways in the database.

### 5.13. Differential Protein Screening

The relative quantitative value of each sample was taken as log 2 (to make the data conform to the normal distribution), and the *p*-value was calculated by *t*-test. When *p* ≤ 0.05, a fold change (FC) change >1.2 was used as the threshold for significant upregulation. FC < 0.67 was taken as the threshold for significant downregulation. Fisher’s exact paired-end test was used to detect the enrichment of DEPs for all identified proteins, and *p* ≤ 0.05 was considered statistically significant. The DEPs were considered significantly enriched at this time.

### 5.14. Statistical Analysis

The data are expressed as the mean ± S.E.M. All statistical analysis was performed with SAS 9.4 statistical software. One-way analysis of variance (ANOVA) was applied to analyze differences in data for the biochemical parameters among the different groups, followed by Dunnett’s significant post hoc test for pair-wise multiple comparisons. *p* < 0.05 indicates a significant difference, and *p* < 0.01 indicates an extremely significant difference. *p* > 0.05 indicates an insignificant difference. GraphPad Prism 8.0.1 was used to create graphics.

## Figures and Tables

**Figure 1 ijms-24-05793-f001:**
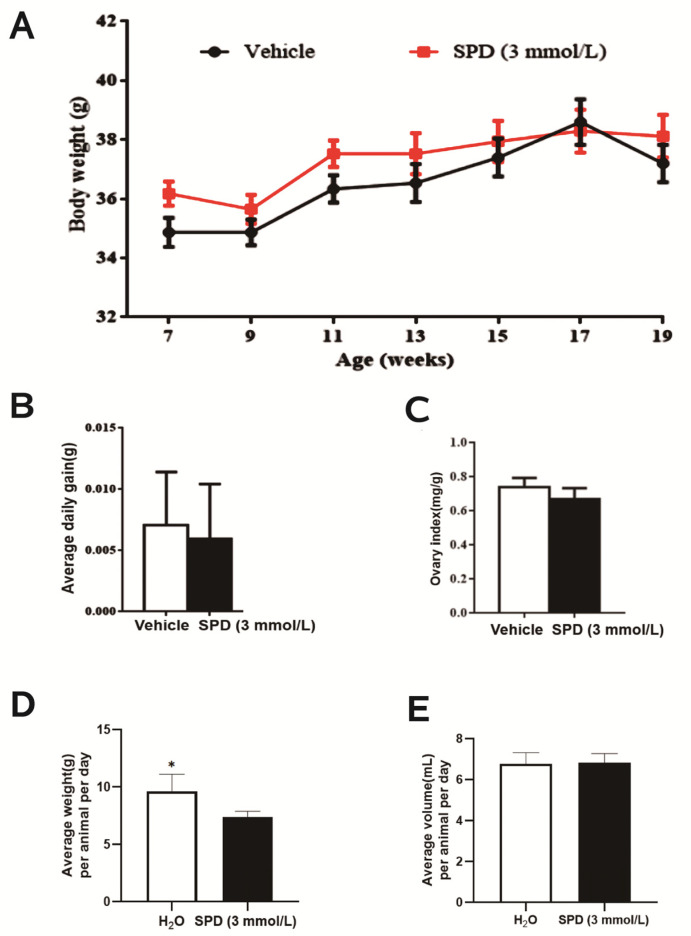
The effect of spermidine on water intake, body weight and ovarian index in mice. (**A**) Weight (g) every two weeks. (**B**) Average daily gain (g). (**C**) Ovarian index (ovarian index = ovarian wet weight (mg)/body weight (g)). (**D**) Food uptake (g). (**E**) Water uptake (mL). * *p* < 0.05 vs. control.

**Figure 2 ijms-24-05793-f002:**
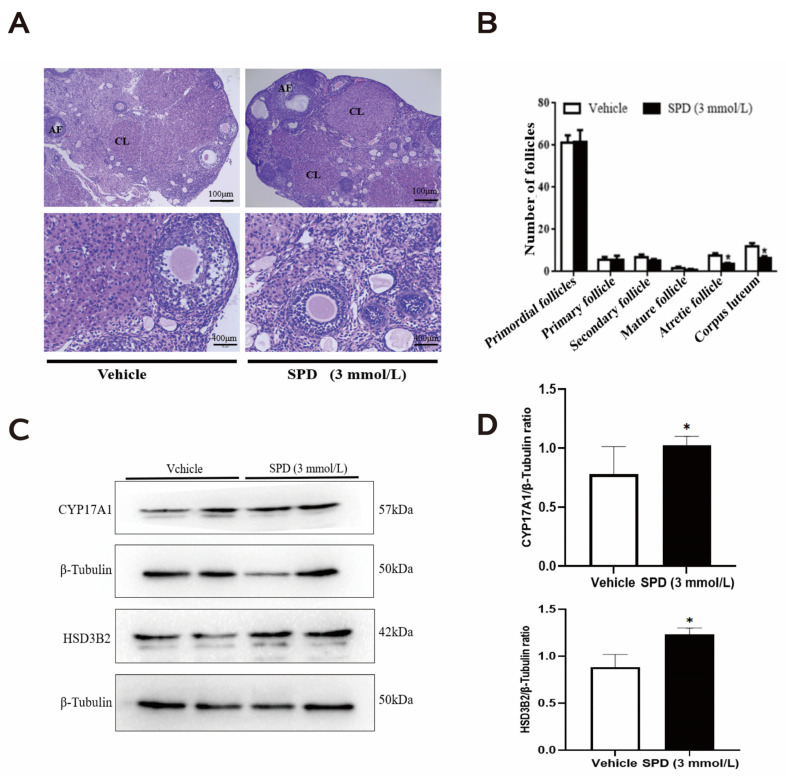
The effects of spermidine on ovarian histomorphology and follicular development in mice. (**A**) Hematoxylin and eosin (HE) staining. (**B**) Graph of the number of follicles in different developmental stages (AF: atresia follicle; CL: corpus luteum). (**C**) Western blot visualizing the expression levels of follicular development-related proteins. (**D**) Quantitative expression levels of follicular development-related proteins. The results shown are representative of three independent experiments. * *p* < 0.05 vs. control.

**Figure 3 ijms-24-05793-f003:**
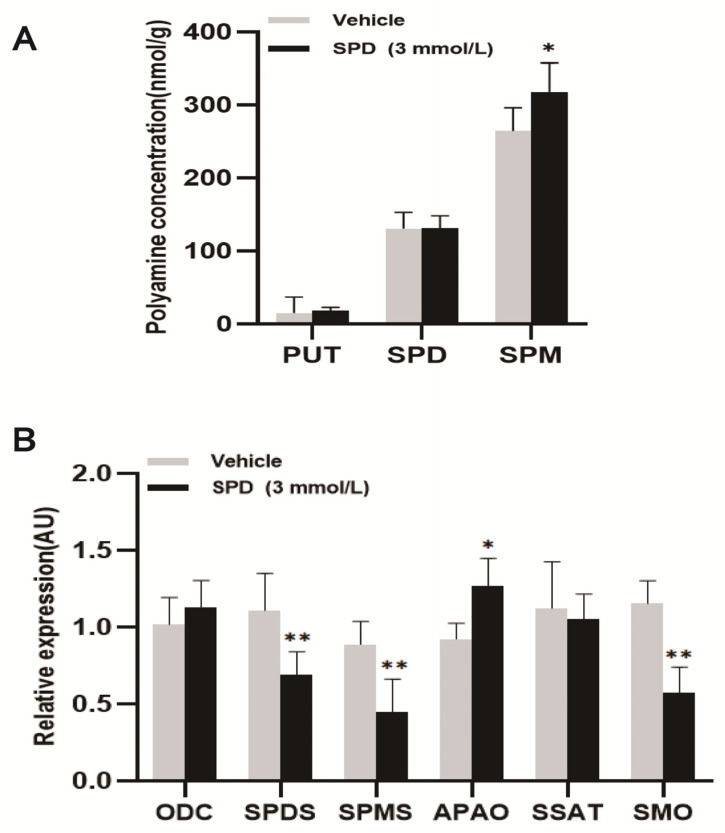
The effects of spermidine on polyamine content and the expression of key metabolic genes in mouse ovaries. (**A**) The effect of spermidine on polyamine content in mouse ovary (PUT: putrescine; SPD: spermidine; SPM: spermine (nmol/g)). (**B**) The effects of spermidine on the expression of key genes of polyamine metabolism in the mouse ovary. The results shown are representative of three independent experiments. * *p* < 0.05 vs. control. ** *p* < 0.01 vs. control.

**Figure 4 ijms-24-05793-f004:**
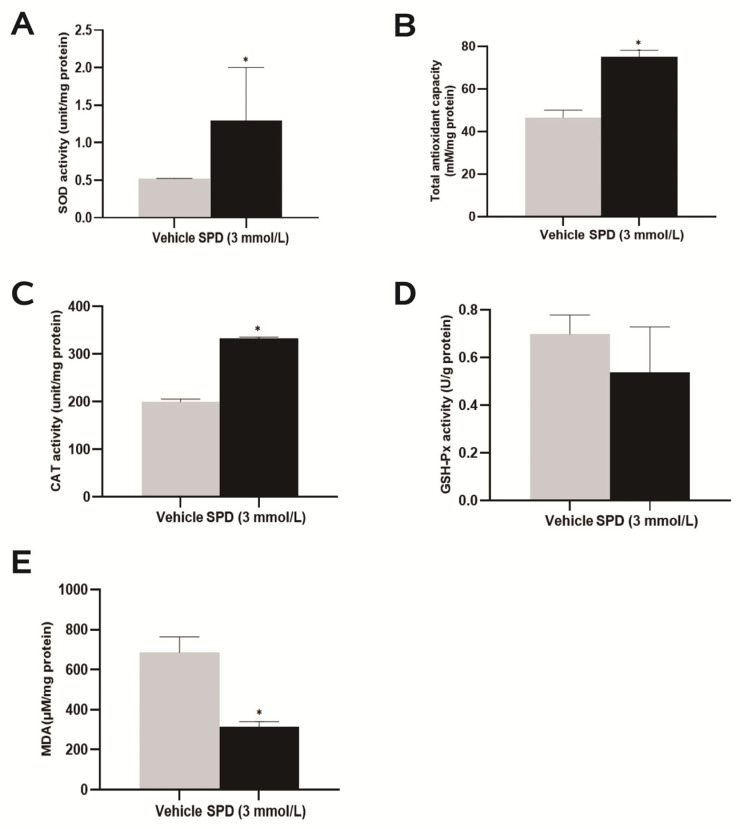
The effects of spermidine on total antioxidant capacity, antioxidant enzymes and lipid peroxidation in mouse ovaries. (**A**) SOD enzyme activity detection. (**B**) Total antioxidant capacity test. (**C**) CAT viability detection. (**D**) GSH-px viability detection. (**E**) MDA level detection. The results shown are representative of three independent experiments. * *p* < 0.05 vs. control.

**Figure 5 ijms-24-05793-f005:**
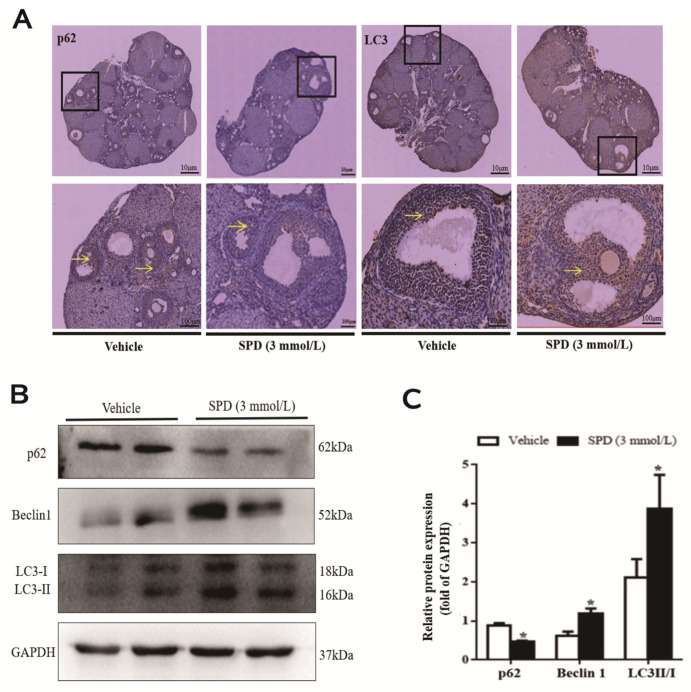
The effect of spermidine on the level of autophagy in mouse ovaries. (**A**) The effect of spermidine on the distribution of autophagy marker proteins determined by immunohistochemistry. Yellow arrow represents the distribution of autophagy-related antigen (cytoplasm). (**B**) Western blot visualization of the expression level of autophagy. (**C**) Quantitative expression levels of autophagy-related proteins. The results shown are representative of three independent experiments. * *p* < 0.05 vs. control.

**Figure 6 ijms-24-05793-f006:**
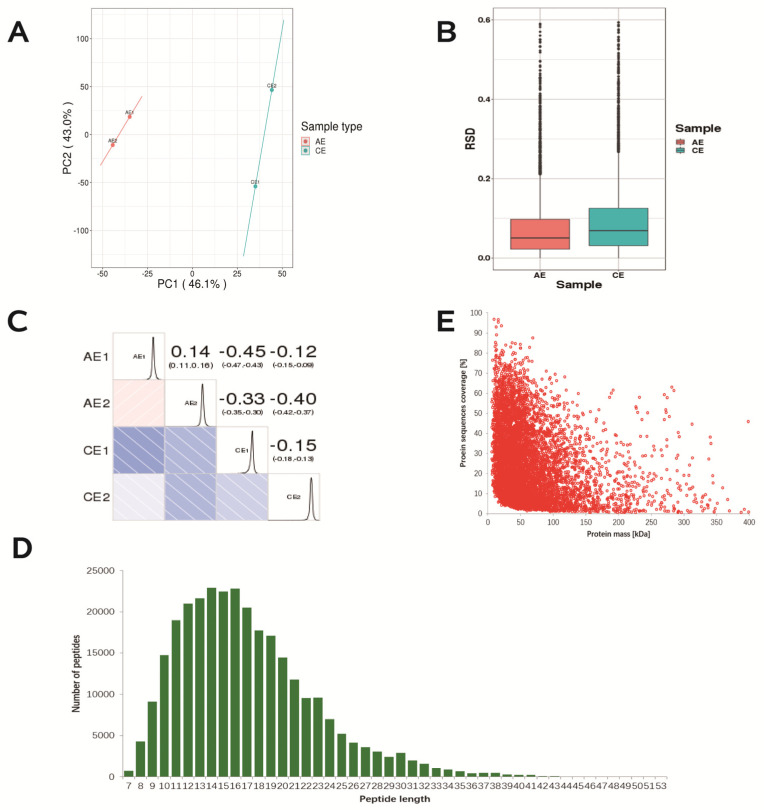
Repeated sample consistency test and identification of differential proteins. (**A**) Two−dimensional scatter plot of protein quantitative principal component analysis (PCA) between replicate samples. (**B**) Boxplot of protein quantification of the relative standard deviation (RSD) distribution between replicates. (**C**) Pearson correlation coefficient (Pearson) heatmap for protein quantification between pairwise samples. (**D**) Length distribution of peptides identified by mass spectrometry. (**E**) Relationship between molecular weight and coverage of proteins identified by mass spectrometry.

**Figure 7 ijms-24-05793-f007:**
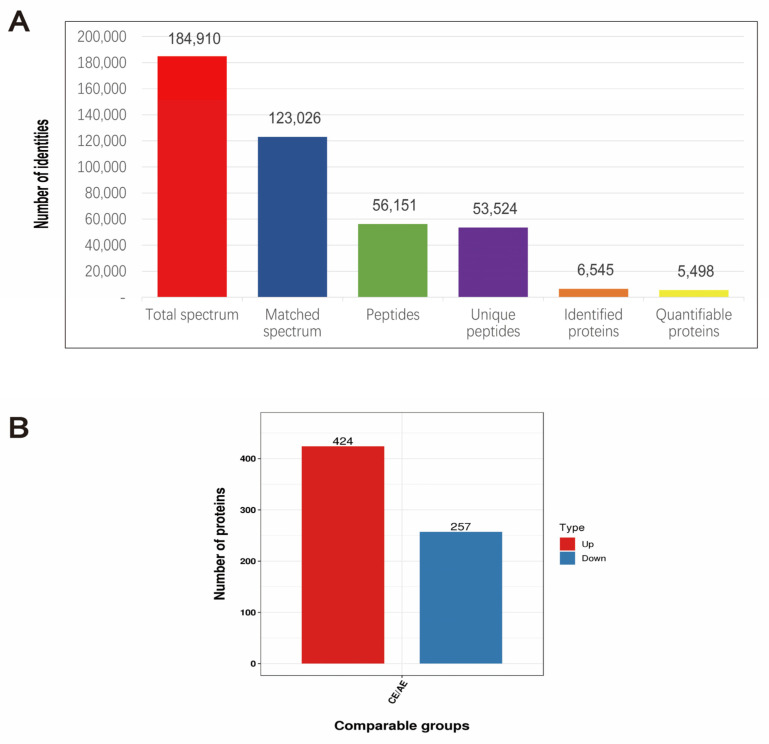
Expression of differentially expressed proteins. (**A**) The basic statistics of mass spectrometry data. (**B**) The numerical distribution of differentially expressed proteins in different comparison groups.

**Figure 8 ijms-24-05793-f008:**
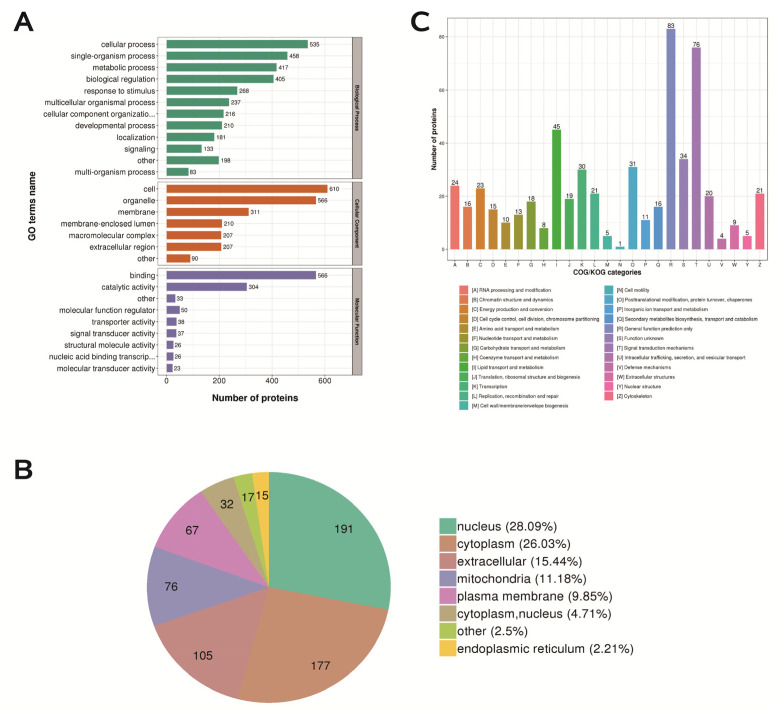
Functional classification, subcellular structure localization classification, and COG/KOG functional classification of differentially expressed proteins. (**A**) Statistical distribution of differentially expressed proteins in GO secondary classification. (**B**) Distribution of subcellular structure localization of differentially expressed proteins. (**C**) Distribution of COG/KOG functional classification of differentially expressed proteins.

**Figure 9 ijms-24-05793-f009:**
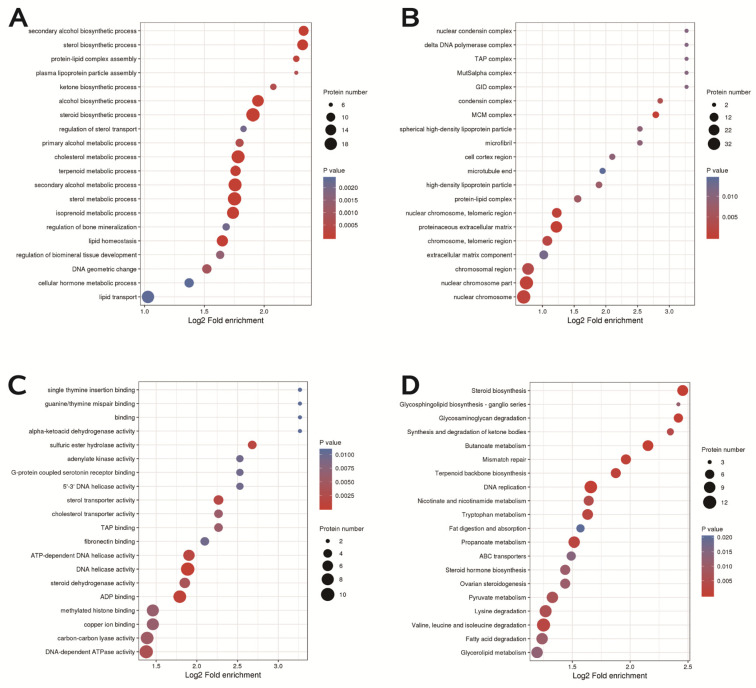
GO enrichment and KEGG pathway enrichment. (**A**) Distribution of biological processes of differentially expressed proteins. (**B**) Distribution of cellular components of differentially expressed proteins. (**C**) Bubble diagram of molecular function distribution of differentially expressed proteins. (**D**) Bubble diagram of the enrichment distribution of differentially expressed proteins in the KEGG pathway.

**Figure 10 ijms-24-05793-f010:**
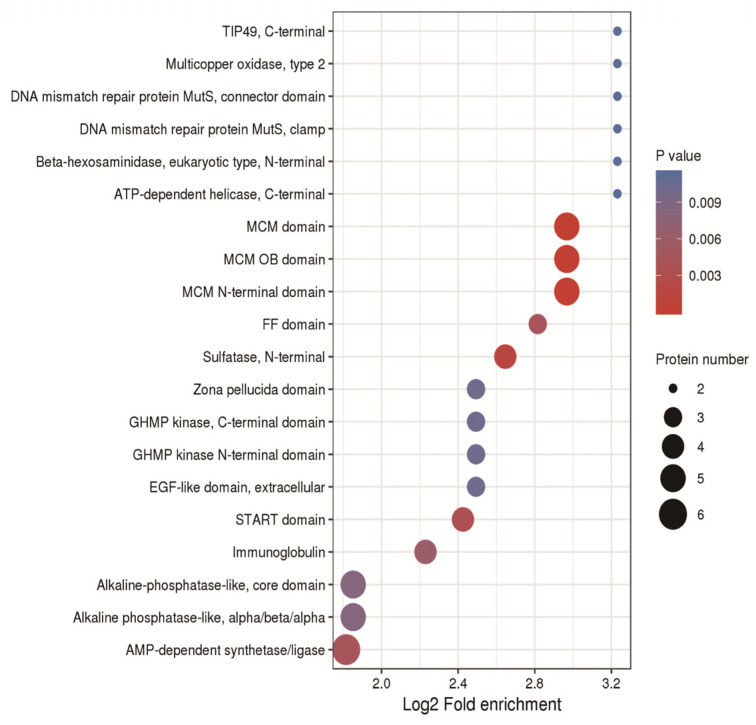
Bubble chart of the enrichment distribution of differentially expressed proteins in protein domain classification.

**Figure 11 ijms-24-05793-f011:**
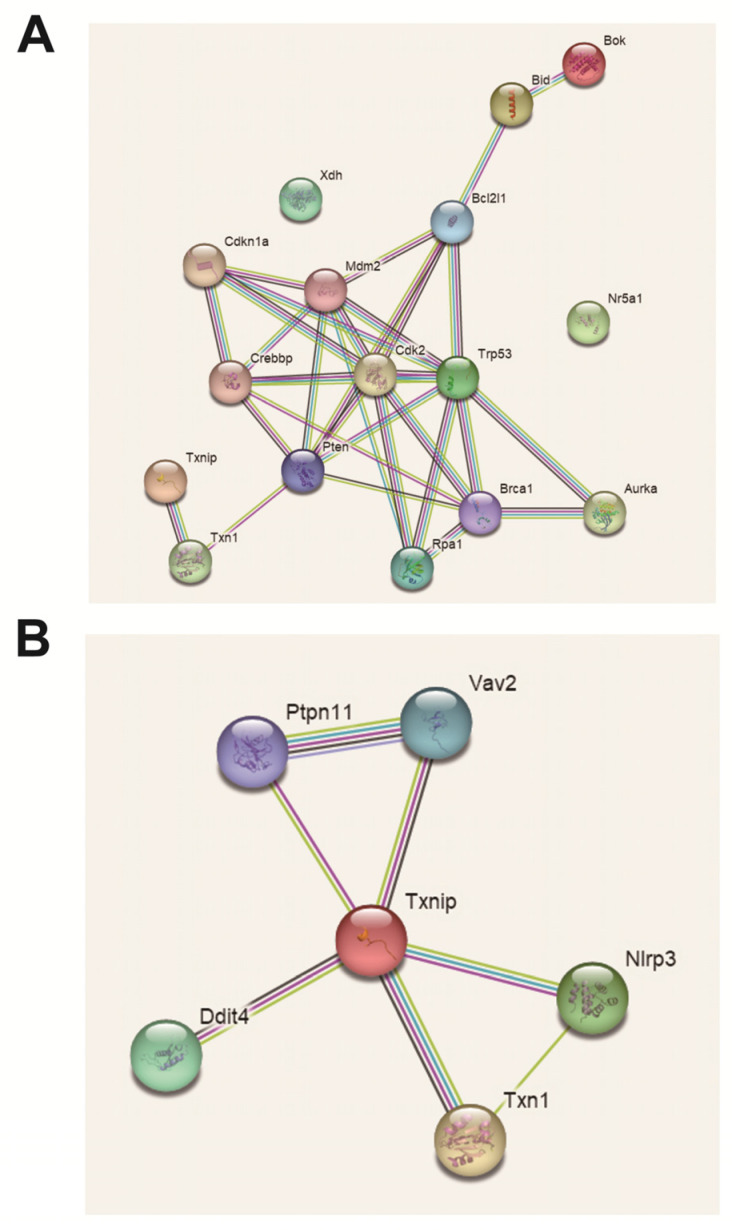
Differentially expressed protein interaction visualization (PPI) map. (**A**) Personalized analysis of differentially expressed protein interactions. (**B**) Personalized analysis of key protein interactions.

**Table 1 ijms-24-05793-t001:** Differentially expressed proteins of spermidine affecting ovarian function in mice.

Serial Number	Protein Name	Expression Regulation	Fold Change
O35425	Bok	Up	1.546
P70444	Bid	Up	1.4635
P33242	Nr5a1	Up	1.955
Q00519	XDH	Down	0.5005
P70399	p53	Up	1.445
Q8BG60	TXNIP	Down	0.5075

**Table 2 ijms-24-05793-t002:** Bioinformatics analysis software.

Bioinformatics Analysis Methods	Tool	Version and URL
Mass spectrum data analysis	MaxQuant	v.1.5.2.8 http://www.maxquant.org/ (accessed on 29 May 2021)
Motif analyze	MoMo	v.5.0.2 http://meme-suite.org/tools/momo (accessed on 29 May 2021)
GO Notes	InterProScan	v.5.14-53.0 http://www.ebi.ac.uk/interpro/ (accessed on 29 May 2021)
Domain Notes	InterProScan	v.5.14-53.0 http://www.ebi.ac.uk/interpro/ (accessed on 29 May 2021)
(KEGG Notes	KAAS	v.2.0 http://www.genome.jp/kaas-bin/kaas_main (accessed on 29 May 2021)
	KEGG Mapper	V2.5 http://www.kegg.jp/kegg/mapper.html (accessed on 29 May 2021)
Subcellular localization	Wolfpsort	v.0.2 http://www.genscript.com/psort/wolf_psort.html (accessed on 29 May 2021)
	CELLO	v.2.5 http://cello.life.nctu.edu.tw/
Enrichment analysis	Perl module	v.1.31 https://metacpan.org/pod/Text::NSP::Measures::2D::Fisher (accessed on 29 May 2021)
Cluster heatmap	R Package pheatmap	v.2.0.3 https://cran.r-project.org/web/packages/cluster/ (accessed on 29 May 2021)
Protein interaction	Blast	v.2.2.26 http://blast.ncbi.nlm.nih.gov/Blast.cgi (accessed on 29 May 2021)
	R package network D3	v.0.4 https://cran.r-project.org/web/packages/networkD3/ (accessed on 29 May 2021)

**Table 3 ijms-24-05793-t003:** Primer sequences used in q RT-PCR.

Gene	Primer Sequence (5’-3’)	Gene Accession Number	Tm (°C)	Amplified Fragment (Bp)
*β-actin*	F: GGGTCAGAAGGACTCCTATG R: GTAACAATGCCATGTTCAAT	XM_015141809.2	57.0	90
*ODC*	F: TTGACTGCCACATCCTTG R: GCTCTGCTATCGTTACACT	XM_021201619.1	58.0	199
*SPDS*	F: ACCAGCTCATGAAGACAGCACTCA R: TGCTACACAGCATGAAGCCGATCT	XM_021160349.1	60.0	189
*SPMS*	F: TTCGGGTGACTCAGTTCCTGCTAA R: AACGGAGACCCTCCTTCAGCAAAT	XM_009214.4	60.0	199
*APAO*	F: AGTCTTCACATGTGCTCTGTGGGT R: TGGCAATTGTGGGTTTCCTGTCAC	XM_021167504.1	59.0	131
*SSAT*	F: TGCCGGTGTAGACAATGACAACCT R: TAAAGCTTTGGAATGGGTGCTCGC	XM_021153071.1	59.0	114
*SMO*	F: TVTGCACAGAGATGCTTCGACAGT R: TTGAGCCCACCTGTGTGTAGGAAT	XM_021184579.1	59.0	129

## Data Availability

The raw data of this proteomics-seq has been uploaded to the iProX database. The data access link in iProX: https://www.iprox.cn/page/project.html?id=IPX0005545000.(accessed on 29 May 2021).

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
