# Peer review of "Exploration of the Antioxidant Effect of Spermidine on the Ovary and Screening and Identification of Differentially Expressed Proteins"

_ijms, 2023, doi:10.3390/ijms24065793_

Round 1
Reviewer 1 Report
In this study, the authors have explored the antioxidant effect of spermidine on the ovary of mice and showed that the number of atretic follicles in the ovaries of spermidine-treated mice was significantly lower than that in the control group. I have the following recommendations: 1) in the Introduction part the authors should clearly determine the main goal of their research (may be like in the Title). 2) in addition, it seems that literature data from Introduction and Discussion parts are crossed and repeated some times, so the authors should approach with criticism once more. 3) In the abstract the details of experiments could be omitted, but the main conclusions should be added.
Author Response
Response to Reviewer 1 Comments
In this study, the authors have explored the antioxidant effect of spermidine on the ovary of mice and showed that the number of atretic follicles in the ovaries of spermidine-treated mice was significantly lower than that in the control group. I have the following recommendations:
Point 1: in the Introduction part the authors should clearly determine the main goal of their research (may be like in the Title).
Response 1: Thank you very much for your suggestions. We accept this opinion and have revised. See line number 107-115.
Point 2: in addition, it seems that literature data from Introduction and Discussion parts are crossed and repeated some times, so the authors should approach with criticism once more.
Response 2: We accept this opinion and have revised. See line number 49-51, 364-367, 343-344.
Point 3: In the abstract the details of experiments could be omitted, but the main conclusions should be added.
Response 3: We accept this opinion have modified it. See line number 15-29.
Spermidine is a naturally occurring polyamine compound that has many biological functions, such as inducing autophagy and anti-inflammatory and anti-aging effects. Spermidine can affect follicular development and thus protect ovarian function. In this study, ICR mice were fed ex-ogenous spermidine drinking water for three months to explore the regulation of ovarian func-tion by spermidine. The results showed that the number of atretic follicles in the ovaries of spermidine-treated mice was significantly lower than that in the control group. Antioxidant en-zyme activities (SOD, CAT, T-AOC) significantly increased and MDA levels significantly de-creased. The expression of autophagy protein (Beclin-1 and microtubule-associated protein 1 light chain 3 LC3 II/I) significantly increased and the expression of the polyubiquitin-binding protein p62/SQSTM 1 significantly decreased. Moreover, we found 424 differentially expressed proteins (DEPs) were upregulated and 257 were downregulated using proteomic sequencing. Gene On-tology and KEGG analyses showed that these DEPs were mainly involved in lipid metabolism, oxidative metabolism and hormone production pathways. In conclusion, spermidine protects the ovarian function by reducing the number of atresia follicles, regulating the level of autoph-agy protein, antioxidant enzyme activity and pol-yamine metabolism in mice.
Reviewer 2 Report
The work entitled “Exploration of the antioxidant effect of spermidine on the ovary and screening and identification of differentially expressed proteins” provides interesting insights into the plausible modulatory roles of spermidine in the ovary physiology, pointing to autophagy and oxidative stress as the cornerstones in this regulatory capacity. Moreover, authors point to several molecular targets that could be implicated in this process, opening the window to future works in the field. Nevertheless, the manuscript presents several flaws, and an in-depth revision of its writing and format is required (nevertheless, discussion is of good quality and easy to read). Some examples are listed below:
Abstract
-Abstract should be improved, introducing the importance of studying ovarian physiology and highlighting the relevance of the results obtained.
-“The autophagy levels (Beclin-1 and microtubule-associated protein 1 light chain 3 LC3 II/I) significantly increased”: does autophagic process really increase, or just protein expression? If so, rewrite.
-In the last sentence of the abstract, authors affirm to have studied some proteins, what kind of analysis, which were the results obtained?. Moreover, it would facilitate the reader if author specify those proteins to be part of the DEPs pool.
-Abstract lack of a concluding sentence.
-Most of the text in the graphical abstract is hard to read.
Introduction
-Enlarge the information about ovarian physiology, introducing the biology and importance of follicular atresia.
-“Apoptosis of granulosa cells is the initiating factor of follicular atresia, oxidative stress induces apoptosis and spermidine is an antioxidant[16]. Follicular atresia is not conducive to the improvement of animal productivity and is also associated with premature ovarian failure[3]; spermidine is closely related to ovarian function” : rewrite to make it clearer. Moreover, in my opinion, these sentences would be more useful connecting both ovarian physiology and polyamines sections.
Results
-When trends are observed, but no statistical significance is reached, instead of typing (p>0.05) indicate the exact p-value. Moreover, if the methods section indicates 0.05 to be considered as the limit for statistical differences, there is no need to specify it again in the text in every comparison.
-Change “The daily feed intake increased significantly” for “The daily food intake increased significantly”.
-Figure descriptions should be self-explanatory, so abbreviations should be included, even if they have been previously indicated in the text.
-Figure format along the manuscript should be improved and constant: maintain a single color pattern for every group under study, and use either lateral legend or name groups under corresponding bars, but not both. Lateral legend in Figure 1B cannot be completely seen.
-Change “To verify the effect of spermidine on mouse follicular development, we detected proteases related to” for “To verify the effect of spermidine on mouse follicular development, we quantified proteases related to”.
-Figure 2B: change “Quantity of follicles” for “Number of follicles”.
-Text relative to Figure 3A is confusing: spermidine contents are described to be both higher and unchanged, result of a writing mistake I guess (correct also Figure 3A legend in needed).
-In Figure 3B, “**” are placed above the bar corresponding to the control group, instead of the treatment group.
-Sometimes, abbreviations appear in the text without being introduced (i.e., T-AOC). Please, pay attention.
-Rewrite: “Both SOD and CAT enzyme activities were significantly higher than those of the control group (p < 0.05), and they were 2.50 times and 1.67 times those DEPs of the control group, respectively (Figure 4A, C).”
-Change “SPD antioxidant stress is achieved through the rescue of autophagic flux” for “SPD antioxidant effect is achieved through the rescue of autophagic flux”.
-Figure 6: again, the figure description lack of required information (i.e., abbreviations).
-In the section 2.7, authors affirm to have screened 6545 differentially expressed proteins, but a fey lines later, only 681 are indicated. Maybe, in the first place, authors refer to the total amount of proteins studied.
-Figure 7A lack of axis title.
-Table 1 is confusing, why authors choose to specify only those proteins in the table? Table title is also confusing.
-Sections 2.8 and 2.9 are redundant. They should be unified and simplified.
-Text in section 2.10 is more likely to be a discussion of the results. Moreover, section 2.10 could also be included as part of a bigger section aimed at characterizing the function, localization and structure of DEPs in the study.
-Again, text in section 2.10 is more likely to be a discussion of the results.
Discussion
-In the sentence: “In this experiment, the expression of SMO in the ovary decreased significantly after spermidine feeding. Our experimental results are consistent with previous results”, which results are referring the authors to? Is there any citation missing?
-“It is speculated that exogenous spermidine can regulate polyamine catabolism by inhibiting the expression of SMO…”. If this hypothesis is extracted from the results rewrite to make it clear.
-“Our results are consistent with previous findings that feeding with 3 mmol·L-1 spermidine significantly increased the T-AOC/SOD and CAT” and “Consistent with existing results, we found that spermidine also induces autophagy in mouse ovaries”, again, citations missing?
-“The levels of T, LH and E2 the mRNA levels of TXNIP and IGF-1…”, use of abbreviations not explained previously.
-Discussion is well written, but I miss a greater effort in highlighting the biological importance of the findings here reported, and how these findings could impact the society.
Conclusions
-Conclusions are poor and do not reflect properly the results previously discussed.
Methodology
-Ovarian index calculation should be included in methodology.
-“The tools and websites used for the analysis are shown in Table 1”, wrong table code.
-“The ovarian tissue was fixed in 4% paraformaldehyde for more than 24 h”, a more concrete fixing time is required.
-“After 5 min, the heating was repeated once”, unclear.
-Rewrite “The cells were incubated at 37 ℃, on a shaking table for 1 hour…”.
-Methodology lack a detailed “Statistical analysis” section, in which some of the information provided in the section 5.9 (“The data were analyzed by one-way ANOVA and multiple comparisons…”) must be individualized and completed.
References
-Some references are numbered twice.
Author Response
Response to Reviewer 2 Comments:
The work entitled “Exploration of the antioxidant effect of spermidine on the ovary and screening and identification of differentially expressed proteins” provides interesting insights into the plausible modulatory roles of spermidine in the ovary physiology, pointing to autophagy and oxidative stress as the cornerstones in this regulatory capacity. Moreover, authors point to several molecular targets that could be implicated in this process, opening the window to future works in the field. Nevertheless, the manuscript presents several flaws, and an in-depth revision of its writing and format is required (nevertheless, discussion is of good quality and easy to read). Some examples are listed below:
Point 1: Abstract
1)-Abstract should be improved, introducing the importance of studying ovarian physiology and highlighting the relevance of the results obtained.
Response 1)- Response 1: Thank you very much for your suggestions. We accept the opinion and have revised it. See line number 15-17.
2)-“The autophagy levels (Beclin-1 and microtubule-associated protein 1 light chain 3 LC3 II/I) significantly increased”: does autophagic process really increase, or just protein expression? If so, rewrite.
Response 2)- Done. See line number 22.
3)-In the last sentence of the abstract, authors affirm to have studied some proteins, what kind of analysis, which were the results obtained?. Moreover, it would facilitate the reader if author specify those proteins to be part of the DEPs pool.
4)-Abstract lack of a concluding sentence.
Response 3) 4)- Done. See line number 27-29.
5)-Most of the text in the graphical abstract is hard to read.
Response 5)- Done.
Point 2: Introduction
1)-Enlarge the information about ovarian physiology, introducing the biology and importance of follicular atresia.
Response 1)- Done. See line number 34-36,39-44.
2)-“Apoptosis of granulosa cells is the initiating factor of follicular atresia, oxidative stress induces apoptosis and spermidine is an antioxidant[16]. Follicular atresia is not conducive to the improvement of animal productivity and is also associated with premature ovarian failure[3]; spermidine is closely related to ovarian function” : rewrite to make it clearer. Moreover, in my opinion, these sentences would be more useful connecting both ovarian physiology and polyamines sections.
Response 2)- Done. See line number 60-63.
Point 3: Results
1)-When trends are observed, but no statistical significance is reached, instead of typing (p>0.05) indicate the exact p-value. Moreover, if the methods section indicates 0.05 to be considered as the limit for statistical differences, there is no need to specify it again in the text in every comparison.
2)-Change “The daily feed intake increased significantly” for “The daily food intake increased significantly”.
Response 1) 2)- Done. See line number 117-122.
3)-Figure descriptions should be self-explanatory, so abbreviations should be included, even if they have been previously indicated in the text.
4)-Figure format along the manuscript should be improved and constant: maintain a single color pattern for every group under study, and use either lateral legend or name groups under corresponding bars, but not both. Lateral legend in Figure 1B cannot be completely seen.
Response 3) 4)- Done.
5)-Change “To verify the effect of spermidine on mouse follicular development, we detected proteases related to” for “To verify the effect of spermidine on mouse follicular development, we quantified proteases related to”.
Response 5)- Done. See line number 136-137.
6)-Figure 2B: change “Quantity of follicles” for “Number of follicles”.
Response 6) - Done.
7)-Text relative to Figure 3A is confusing: spermidine contents are described to be both higher and unchanged, result of a writing mistake I guess (correct also Figure 3A legend in needed).
8)-In Figure 3B, “**” are placed above the bar corresponding to the control group, instead of the treatment group.
Response 7)8)- Thank you very much for your suggestions. We accept this opinion and have revised the figure. See line number 154.
9)-Sometimes, abbreviations appear in the text without being introduced (i.e., T-AOC). Please, pay attention.
10)-Rewrite: “Both SOD and CAT enzyme activities were significantly higher than those of the control group (p < 0.05), and they were 2.50 times and 1.67 times those DEPs of the control group, respectively (Figure 4A, C).”
Response 9)10)- Thank you very much for your suggestions. We accept this opinion and have revised it. See line number 172-176.
11)-Change “SPD antioxidant stress is achieved through the rescue of autophagic flux” for “SPD antioxidant effect is achieved through the rescue of autophagic flux”.
Response 11)- Done. See line number 187.
12)-Figure 6: again, the figure description lack of required information (i.e., abbreviations).
Response 12)- Thank you very much for your suggestions. We have revised it. See line number 223-226.
13)-In the section 2.7, authors affirm to have screened 6545 differentially expressed proteins, but a fey lines later, only 681 are indicated. Maybe, in the first place, authors refer to the total amount of proteins studied.
Response 13)- Thank you very much for your suggestions. There are four kinds of fold change (>1.2, 1.3, 1.5 and 2). When testing the repeatability of samples, the repeatability of 1.2 is better. So we select proteins with fold change>1.2 in total protein in Method 5.13 When screening differential proteins.
14)-Figure 7A lack of axis title.
Response 14) - Thank you very much for your suggestions. We have revised the figure.
15)-Table 1 is confusing, why authors choose to specify only those proteins in the table? Table title is also confusing.
Response 15)- Thank you very much for your suggestions. The purpose of this article is to screen for proteins related to ovarian function, spermidine-induced autophagy antioxidation, so only these related proteins are selected. Table 1. Differentially expressed proteins of spermidine affecting ovarian function in mice
16)-Sections 2.8 and 2.9 are redundant. They should be unified and simplified.
Response 16)- Thank you very much for your suggestions. We accept this opinion and simplified it. See line number 249-253, 262-271.
17)-Text in section 2.10 is more likely to be a discussion of the results. Moreover, section 2.10 could also be included as part of a bigger section aimed at characterizing the function, localization and structure of DEPs in the study.
-Again, text in section 2.10 is more likely to be a discussion of the results.
Response 17)- Thank you very much for your suggestions. We have added some to the discussion. See line number 280-283, 405-413.
Point 3: Discussion
1)-In the sentence: “In this experiment, the expression of SMO in the ovary decreased significantly after spermidine feeding. Our experimental results are consistent with previous results”, which results are referring the authors to? Is there any citation missing?
2)-“It is speculated that exogenous spermidine can regulate polyamine catabolism by inhibiting the expression of SMO…”. If this hypothesis is extracted from the results rewrite to make it clear.
3)-“Our results are consistent with previous findings that feeding with 3 mmol·L-1 spermidine significantly increased the T-AOC/SOD and CAT” and “Consistent with existing results, we found that spermidine also induces autophagy in mouse ovaries”, again, citations missing?
Response 1)2)3)- Thank you very much for your suggestions. We have inserted references. See 325-328, 353-356.
4)-“The levels of T, LH and E2 the mRNA levels of TXNIP and IGF-1…”, use of abbreviations not explained previously.
Response 4)- Thank you very much for your suggestions. We have revised it. See line number 439-442.
5)-Discussion is well written, but I miss a greater effort in highlighting the biological importance of the findings here reported, and how these findings could impact the society.
Response 5)- Thank you very much for your suggestions. We have revised it. See line number 457-460.
Point 4: Conclusions
-Conclusions are poor and do not reflect properly the results previously discussed.
Response 4- Thank you very much for your suggestions. The conclusion has been rewrited.
See line number 462-470.
Point 5: Methodology
1)-Ovarian index calculation should be included in methodology.
Response 1)- It has been inserted. See line number 483-485.
2)-“The tools and websites used for the analysis are shown in Table 1”, wrong table code.
Response 2)- Done.See in line number 504.
3)-“The ovarian tissue was fixed in 4% paraformaldehyde for more than 24 h”, a more concrete fixing time is required.
Response 3)- It has been revised. See line number 506.
4)-“After 5 min, the heating was repeated once”, unclear.
Response 4)- Done. See line number 517-518.
5)-Rewrite “The cells were incubated at 37 ℃, on a shaking table for 1 hour…”.
Response 5)- Thank you very much for your suggestions. I accept this opinion. See line number 581-584.
6)-Methodology lack a detailed “Statistical analysis” section, in which some of the information provided in the section 5.9 (“The data were analyzed by one-way ANOVA and multiple comparisons…”) must be individualized and completed.
Response 6)- Done. See line number 623-630.
Point 6: References
-Some references are numbered twice.
Response 6- References have been corrected.